# Risk factors for emergent delivery before 36 weeks among pregnant women with placenta accreta spectrum disorder

Nawaporn Phetrat, Savitree Pranpanus, Thitima Suntharasaj, Chusana Petpichetchian *

Department of Obstetrics and Gynecology, Faculty of Medicine, Prince of Songkla University, Songkhla, Thailand

* chusana020@gmail.com

## Abstract

Recent studies evaluating risk factors for emergent delivery in women with placenta accreta spectrum disorders have yielded insufficient results. A limited number of studies have evaluated prenatal ultrasound signs of the placenta accreta spectrum as risk factors and have reported inconsistent outcomes. This retrospective study included women with suspected prenatal placenta accreta spectrum who delivered between January 2007 and December 2022 at a tertiary hospital in Southern Thailand. Women who delivered electively or for conditions unrelated to the placenta accreta spectrum before 36 weeks of gestation were excluded. Women who underwent emergent delivery before 36 weeks and delivery after 36 weeks were compared using univariate and multivariable analyses. Overall, 174 women with placenta accreta spectrum were included; 45 (25.0%) underwent emergent delivery before 36 weeks of gestation. Women who delivered before 36 weeks had significantly more premature uterine contractions (41.7% vs. 7.0%, $P<0.001$), premature rupture of membranes (8.3% vs. 0%, $P<0.05$), antepartum hemorrhage (75.0% vs. 27.9%, $P<0.001$), and sonographic findings of placental bulging (45.8% vs. 23.3%, $P=0.003$) than those who delivered after 36 weeks. The number of premature uterine contractions and antepartum hemorrhage episodes ($P<0.001$) and more severe placenta accreta spectrum ($P=0.003$) were significantly associated with emergent delivery. Significant predictors of emergent delivery before 36 weeks were a history of preterm birth (odds ratio: 10.1, 95% confidence interval: 1.0–97.4), presence of premature uterine contractions (9.7, 2.9–31.5), antepartum hemorrhage (6.4, 2.2–18.3), severe placenta accreta spectrum (6.1, 1.5–25.0), and placental bulging (4.4, 1.3–14.2). In conclusion, the significant predictors of emergent delivery before 36 weeks of gestation among women with placenta accreta spectrum were a history of preterm birth, premature uterine contractions or antepartum hemorrhage before 34 weeks, placental bulging, and prenatal diagnosis of severe placenta accreta spectrum.

**Data availability statement:** All relevant data are within the manuscript and its Supporting Information files.

**Funding:** The author(s) received no specific funding for this work.

**Competing interests:** The authors have declared that no competing interests exist.

## Introduction

Placenta accreta spectrum (PAS) disorder is a clinical condition in which the placenta does not detach spontaneously after fetal delivery. This is caused by the loss of Nitabuch's layer, the spongiosus layer of the decidua, which allows direct trophoblast invasion into the myometrium. This abnormal implantation prevents normal placental separation, and forced placental removal can cause massive and potentially life-threatening bleeding [1,2]. The incidence of PAS reportedly ranges from 0.79 to 3.11 per 1,000 births after prior cesarean delivery [2], is rising worldwide, and correlates with the cesarean section (CS) rate. In our institution, a tertiary-level hospital in Southern Thailand, the incidence of PAS increased from 20.1 to 62.0 per 10,000 deliveries between 2008 and 2018 [3]. PAS is one of the most dangerous pregnancy conditions because it is significantly associated with maternal morbidity and mortality [4].

Prenatal diagnosis followed by scheduled CS by an experienced team has been shown to improve maternal outcomes. Emergent delivery has been reported to be associated with maternal morbidities, including intensive care unit (ICU) admission, longer operative time, need for transfusion, higher number of transfused blood units, and longer hospital stay [5–9]. Adverse neonatal outcomes also increase with emergent delivery, such as low neonatal birth weight, need for mechanical ventilation, and longer stay in the neonatal ICU [5,6,9,10]. To avoid unplanned emergent delivery, CS is usually performed before term gestation; however, the benefits of planned delivery must be weighed against the risk of iatrogenic prematurity.

A decision analysis in 2010 recommended that the optimal timing of delivery for women with placenta previa and ultrasonography-based suspicion of PAS is 34 weeks of gestational age (GA). However, this was based on the data of 400 women with placenta previa alone and not placenta previa with PAS [11]. The American College of Obstetrics and Gynecology's recent committee opinion recommended the timing of delivery for women with suspected placenta accreta, increta, or percreta at $34^{+0}$ to $35^{+6}$ weeks GA [12,13]. The International Federation of Gynecology and Obstetrics recommends scheduled non-emergent delivery at 35–37 weeks [14]. Another recommendation by the International Society for Placenta Accreta Spectrum (IS-PAS) suggests that delivery timing should be adjusted for each unique set of circumstances based on the individual woman's risk of emergent delivery [2]. The recommendation concluded that expectant management until after $36^{+0}$ weeks GA is reasonable for women with no history of preterm delivery, significant vaginal bleeding, preterm prelabor rupture of the membrane (PPROM), or preterm labor. In contrast, for women with a history of previous preterm birth, multiple episodes of non-notable vaginal bleeding, or one episode of significant vaginal bleeding or PPROM, planned delivery should be considered at around $34^{+0}$ weeks GA [2].

According to IS-PAS recommendations [2], maternal characteristics can be used to select women who can deliver before or after 36 weeks GA. Studies have shown that factors associated with emergent CS in women with PAS are maternal body mass index (BMI), hypertensive disorders, intrahepatic cholestasis, number of prior

CS, antepartum hemorrhage (APH), multiple episodes of bleeding, history of preterm labor, PPROM, and severity of PAS [6,7,9,15–18]. However, to date, these results have not been consistently reported. Most of these studies had a retrospective design with small sample sizes, and some reported a population comprising a mixture of patients with PAS and placenta previa. Some of the risk factors assessed in previous studies were clinical symptoms that occurred throughout gestation but not within a specific GA range. The outcome of interest was emergent delivery at any given GA. It would be difficult to apply this knowledge prospectively in clinical practice, where delivery timing must be planned at approximately 34 weeks GA, regarding whether to perform elective delivery before 36 weeks or extend the pregnancy beyond 36 weeks. In addition, only two recent studies have assessed the use of prenatal ultrasound (US) signs for PAS diagnosis to predict emergent delivery [7,17]. The results of these studies were not significant, and their sample sizes were relatively small. Therefore, we conducted this study to evaluate the risk factors associated with emergent delivery before 36 weeks GA in women with PAS, including maternal characteristics and ultrasound signs before 34 weeks GA.

## Materials and methods

This retrospective cohort study was conducted at a tertiary hospital in Southern Thailand. Women with singleton pregnancies and prenatal suspicion of PAS disorder who delivered at the study hospital after 24 weeks GA between January 2007 and December 2022 were included. Women who delivered before 36 weeks GA because of conditions unrelated to PAS, such as maternal hypertensive disorders or fetal growth restriction, and those who underwent elective delivery at 34–35$^{+6}$ weeks GA were excluded.

After the institutional review board approved the study protocol (REC.65-074-12-4), the medical records of patients who were evaluated for prenatal suspicion of PAS disorder during the study period were reviewed between April 4, 2022, and April 3, 2023. Information that could identify individual participants was their hospital number, and this information was available only to the first author (NP) and corresponding author (CP). The need for consent was waived because of the retrospective study design. Clinical data, including maternal and obstetrical characteristics and perinatal and postnatal outcomes, were extracted from the electronic medical database by the first author (NP). APH was defined as uterine bleeding that occurred between 14 and 34 weeks GA and required medical attention. Premature uterine contraction (PUC) was defined as a uterine contraction occurring before 34 weeks and requiring hospital admission. US images were reviewed for the presence of each sign of PAS by an experienced maternal-fetal medicine specialist (SP) who was blinded to the final pathological results. Cases of 'suspected PAS' were women with at least one US finding suggestive of PAS according to the descriptors proposed by the European Working Group on Abnormally Invasive Placenta [19]. Patients were then classified in the "emergent delivery before 36 weeks GA" group if delivery was required at <36 weeks GA and prior to the scheduled time for reasons such as active bleeding or active labor. On the other hand, women in whom pregnancy continued beyond 36 weeks GA were classified in the "delivery after 36 weeks GA" group, regardless of the emergent or elective nature of the delivery.

The sample size was determined using the formula for a cohort study based on data from a previous study [6]. The rates of emergent delivery were 19% and 53% among patients without and with APH (p1=0.19, p2=0.53), respectively. The required sample size was 90 patients. Descriptive statistics were reported as mean ± standard deviation or median (interquartile range) for continuous variables and as number and percentage for categorical variables. The chi-squared test, Fisher's exact test, or Student's t-test was used to compare characteristics between delivery groups. Multiple logistic regression analysis was used to identify independent risk factors for emergent delivery before 36 weeks GA. Statistical significance was set at $P<0.05$.

## Results

A total of 227 patients were admitted to our hospital with US-based suspicion of PAS between January 2007 and December 2022. Fifty-three women were excluded from the study because they underwent elective delivery between 34 and 35$^{+6}$ weeks GA, leaving a total number of 174 women for the analysis.

The average maternal age was 34.7 (range: 22–45 years). Nearly all patients (170 women, 97.7%) were parous, and most (165 women, 94.8%) had undergone at least one CS in prior pregnancies. Twelve women (6.9%) had a history of preterm delivery in previous pregnancies. Sixty-nine women (39.7%) experienced APH at 14–34 weeks GA. The average GA at the first APH episode was 27 weeks. At the time of US for PAS diagnosis, the placental locations were anterior in 163 (93.7%) and posterior in 11 (6.3%) women. PAS was suspected in 171 women (98.3%) in the setting of placenta previa or a low-lying placenta. Sixty-four (36.8%), 84 (48.3%), and 26 (14.9%) women had placenta accreta, increta, and percreta, respectively. A diagnosis of PAS was histologically confirmed in 150 patients (86.2%) and was more frequent in women with a prenatal suspicion of placenta percreta (26/26 women, 100%) than in those with placenta increta (77/84 women, 91.7%) or accreta (47/64 women, 73.4%).

Emergent delivery before 36 weeks of GA was required in 45 (25.0%) women (mean GA at delivery, 32$^{+6}$ weeks), whereas pregnancy continued beyond 36 weeks GA in 129 (75.0%) women (mean GA at delivery, 37 weeks). Among women who required emergent delivery, the main indications for delivery were significant bleeding with or without other symptoms in 29 (64.4%) women and 9 (20.0%) women had only PUC without severe bleeding. Table 1 shows a comparison of the demographic and obstetric characteristics between the two groups. Women who required emergent delivery before 36 weeks GA were more likely to have experienced PUC (41.7% vs. 7%, $P<0.001$), PPROM (8.3% vs. 0%, $P<0.05$), and APH (75.0% vs. 27.9%, $P<0.001$) than those in whom pregnancy continued beyond 36 weeks GA. Women

**Table 1. Characteristics of women who underwent emergent delivery before 36 weeks GA vs. delivery after 36 weeks.**

| Characteristic | Emergent delivery <36 weeks, N=45 N (%) | Delivery ≥36 weeks, N=129 N (%) | P value |
|---|---|---|---|
| Age (years)[a] | 34.6±4.4 | 34.7±4.9 | 0.907 |
| BMI (pre-pregnancy)[a] | 26.0±3.7 | 27.1±6.1 | 0.240 |
| Parity | | | 0.387[b] |
| 0 | 0 (0) | 4 (3.1) | |
| 1 | 16 (35.6) | 53 (41.1) | |
| ≥2 | 29 (64.4) | 72 (55.8) | |
| Number of prior CSs | | | 0.921 |
| 0 | 2 (4.4) | 7 (5.4) | |
| 1 | 23 (51.1) | 65 (50.4) | |
| ≥2 | 20 (44.4) | 57 (44.2) | |
| History of preterm birth | 4 (8.9) | 8 (6.2) | 0.461 |
| PUC before 34 weeks GA | 20 (44.4) | 9 (7.0) | **<0.001** |
| Episodes of PUC before 34 weeks | | | **<0.001[b]** |
| - 0 | 25 (55.6) | 120 (93.0) | |
| - 1 | 13 (28.9) | 7 (5.4) | |
| - ≥2 | 7 (15.5) | 2 (1.6) | |
| PPROM before 34 weeks GA | 4 (8.9) | 0 (0) | **0.004[b]** |
| APH between 14 and 34 weeks GA | 33 (73.3) | 36 (27.9) | **<0.001** |
| Number of APH episodes before 34 weeks GA | | | **<0.001** |
| - 0 | 12 (26.7) | 93 (72.1) | |
| - 1 | 10 (22.2) | 20 (15.5) | |
| - ≥2 | 23 (51.1) | 16 (12.4) | |

[a]Data reported as mean±SD, [b]Fisher's exact test.

BMI, body mass index; CS, cesarean section; PUC, premature uterine contraction; PPROM, preterm prelabor rupture of membranes; APH, antepartum hemorrhage; GA, gestational age.

with emergent delivery before 36 weeks GA were also more likely to have experienced more PUC and APH episodes than women whose pregnancy went beyond 36 weeks GA ($P<0.001$). Except for placental bulging, which was significantly more frequent in women who underwent emergent delivery before 36 weeks GA (45.8% vs. 23.3%, $P=0.003$), the prenatal US findings did not differ between the two groups (Table 2). Emergent delivery before 36 weeks GA was associated with a higher degree of PAS ($P=0.003$).

Multiple logistic regression analysis was conducted to identify the factors associated with emergent delivery at <36 weeks GA (Table 3). After adjusting for maternal BMI, history of preterm birth, APH, PUC, and all US markers of PAS, five factors were identified as independent predictors of emergent delivery before 36 weeks GA. These were history of preterm birth (odds ratio [OR] 10.100, 95% confidence interval [CI] 1.048–97.357, $P=0.045$), PUC (OR 9.657, 95% CI 2.962–31.485, $P<0.001$), APH (OR 6.393, 95% CI 2.239–18.256, $P=0.001$), severity of PAS (increta-percreta) (OR 6.148, 95% CI 1.512–25.002, $P=0.011$), and placental bulging (OR 4.351, 95% CI 1.329–14.242, $P=0.015$).

Perinatal outcomes were compared between the two groups (Table 4). Patients who underwent emergent delivery before 36 weeks GA showed significantly higher operative blood loss than those who delivered after 36 weeks GA. The adjacent organ injury rate, ICU admission rate, and operative time were also higher in the emergent delivery group than in the group that delivered after 36 weeks; however, the difference was not significant. The rates of neonatal morbidities, including respiratory distress syndrome (RDS), neonatal sepsis, anemia, jaundice, and low birth weight, were significantly higher among newborns born through emergent delivery before 36 weeks GA than in the group who delivered after 36 weeks.

An additional analysis of the 53 women who had planned delivery between 34 and 35$^{+6}$ weeks GA was performed to provide more details regarding this sub-population. The mean GA at delivery was 34$^{+6}$ weeks. Ten (18.9%) and 29 (54.7%) patients experienced PUC and APH, respectively. Concerning the severity of PAS, 11 (20.7%), 30 (56.6%), and 12 (22.6%) women were prenatally diagnosed with placenta accreta, increta, and percreta, respectively. The mean EBL

**Table 2. Ultrasonographic findings of women who underwent emergent delivery before 36 weeks of gestational age vs. delivery after 36 weeks.**

| US characteristic | Emergent delivery <36 weeks, N=45 N (%) | Delivery ≥36 weeks, N=129 N (%) | P value |
|---|---|---|---|
| Severity of PAS | | | **0.002** |
| Accreta | 7 (15.6) | 57 (44.2) | |
| Increta | 30 (66.7) | 54 (41.9) | |
| Percreta | 8 (17.8) | 18 (13.9) | |
| 2D grayscale | | | |
| Loss of clear zone | 42 (93.3) | 117 (90.7) | 0.588 |
| Abnormal placental lacunae | 39 (86.7) | 98 (76.0) | 0.131 |
| Bladder wall interruption | 29 (64.4) | 72 (55.8) | 0.312 |
| Myometrial thinning | 27 (60.0) | 65 (50.4) | 0.266 |
| Placental bulging | 21 (46.7) | 30 (23.3) | **0.003** |
| Focal exophytic mass | 3 (6.7) | 6 (4.7) | 0.599 |
| 2D color Doppler | | | |
| Uterovesical hypervascularity | 40 (88.9) | 101 (78.3) | 0.119 |
| Sub-placental hypervascularity | 19 (42.2) | 35 (27.1) | 0.060 |
| Bridging vessels | 30 (66.7) | 74 (57.4) | 0.273 |
| Placental lacunae feeder vessels | 29 (64.4) | 63 (48.8) | 0.071 |

US, ultrasonography; PAS, placenta accreta spectrum; US, ultrasonography; 2D, two-dimensional.

**Table 3. Logistic regression analysis evaluating potential predictors of emergent delivery before 36 weeks of gestational age.**

| Characteristic | OR | 95% CI | *P* value |
|---|---|---|---|
| BMI ≥25 kg/m² | 1.227 | 0.479–3.145 | 0.670 |
| History of preterm birth | 10.100 | 1.048–97.357 | **0.045** |
| PUC before 34 weeks GA | 9.657 | 2.962–31.485 | **<0.001** |
| APH between 14 and 34 weeks GA | 6.393 | 2.239–18.256 | **0.001** |
| Increta-percreta | 6.148 | 1.512–25.002 | **0.011** |
| Loss of clear zone | 0.693 | 0.063–2.487 | 0.323 |
| Abnormal placental lacunae | 2.240 | 0.465–10.799 | 0.315 |
| Bladder wall interruption | 0.782 | 0.261–2.342 | 0.660 |
| Myometrial thinning | 0.516 | 0.176–1.518 | 0.230 |
| Placental bulging | 4.351 | 1.329–14.242 | **0.015** |
| Focal exophytic mass | 1.179 | 0.171–8.131 | 0.867 |
| Uterovesical hypervascularity | 0.556 | 0.127–2.441 | 0.437 |
| Sub-placental hypervascularity | 2.344 | 0.792–6.936 | 0.124 |
| Bridging vessels | 0.646 | 0.219–1.908 | 0.429 |
| Placental lacunae feeder | 1.210 | 0.369–3.972 | 0.753 |

OR, odds ratio; CI, confidence interval; BMI, body mass index; PUC, premature uterine contraction; PPROM, preterm prelabor rupture of membranes; APH, antepartum hemorrhage; GA, gestational age.

**Table 4. Maternal and neonatal complications in women who underwent emergent delivery before 36 weeks of gestational age vs. delivery after 36 weeks.**

| Outcome | Emergent delivery <36 weeks, N=45 | Delivery ≥36 weeks, N=129 | *P* value |
|---|---|---|---|
| | N (%) | N (%) | |
| **Maternal complications** | | | |
| Operative time (min)[a] | 216.8±53.4 | 205.0±65.9 | 0.280 |
| Estimated blood loss (mL)[a] | 5082.2±4028.0 | 3665.9±3684.6 | **0.032** |
| Adjacent organ injury | 9 (20.0) | 13 (10.1) | 0.116 |
| Bowel | 1 (2.2) | 3 (2.3) | 1.000[b] |
| Bladder | 8 (17.8) | 11 (8.5) | 0.087 |
| Ureter | 2 (4.4) | 2 (1.6) | 0.275[b] |
| Repeat surgery | 1 (2.2) | 5 (3.9) | 1.000[b] |
| ICU admission | 18 (40.0) | 32 (24.8) | 0.234 |
| Acute kidney injury | 0 (0.0) | 2 (1.6) | 1.000[b] |
| **Neonatal complications** | | | |
| Composite neonatal complications | 38 (84.4) | 46 (35.7) | **<0.001** |
| Respiratory distress syndrome | 27 (60.0) | 16 (12.4) | **<0.001** |
| Transient tachypnea of the newborn | 12 (26.7) | 21 (16.3) | 0.184 |
| Neonatal sepsis | 8 (17.8) | 8 (6.2) | **<0.001** |
| Anemia | 18 (40.0) | 16 (12.4) | **<0.001** |
| Hypoglycemia | 8 (17.8) | 14 (10.9) | 0.296 |
| Jaundice | 22 (48.9) | 20 (15.5) | **<0.001** |
| Low birth weight | 16 (35.6) | 5 (3.9) | **<0.001** |
| Stillbirth | 1 (2.2) | 1 (0.7) | 0.451[b] |

[a]Data are shown as mean±SD, [b]Fisher's exact test.

ICU, intensive care unit; GA, gestational age.

at the time of delivery was 5347.2 (± 3074.2) mL. Composite neonatal complications occurred in 41 (77.4%) newborns, mainly consisting of RDS, transient tachypnea of the newborn, anemia, hypoglycemia, and jaundice in 22 (41.5%), 12 (22.6%), 19 (35.8%), 14 (32.1%), and 18 (34.0%) newborns, respectively.

## Discussion

As emergent delivery in women with PAS is associated with increased maternal morbidities, factors that predict whether the pregnancy will undergo emergent deliveries may play an important role in planning delivery timing. In this study, we found that emergent delivery before 36 weeks GA was required in a quarter of the women with PAS. The risk factors associated with emergent delivery at <36 weeks GA were history of preterm birth in a prior pregnancy, PUC before 34 weeks GA, APH before 34 weeks GA, presence of placental bulging on US, and prenatal diagnosis of severe PAS (placenta increta to percreta). Most of the factors identified in our study, specifically a history of preterm birth, APH, and PUCs, are in concordance with those recommended by the IS-PAS and other studies [1,2,9,10,18]. Regarding PAS severity, Zhao et al. also reported that a more severe PAS type was associated with emergent delivery in a multivariable analysis [18]. This is relevant in clinical practice, as more severe types of PAS pose a greater risk of maternal complications, and a multidisciplinary team consisting of experienced surgeons, anesthesiologists, urologists, and intensivists may be required. Avoiding emergent surgery by timing delivery earlier in such cases seems to be an appropriate approach.

Various US findings have been investigated as possible predictors of emergent delivery in patients with PAS. Fishel Bartal et al. retrospectively evaluated 109 women with PAS and found no significant difference between women who underwent planned and un planned deliveries for sonographic markers, including placental location, placental lacunae, abnormal uterine serosa bladder line, and loss of sonolucency [7]. In the present study, we found no association between these markers and unplanned delivery. Another recent prospective cohort study by the Antenatal Diagnosis of Placental Attachment Disorders study group reported that an interrupted retroplacental hypoechoic space, an interrupted hyperechogenic bladder line, and abnormal placental lacunae were significantly more frequent among women with PAS who required emergent CS [17]. However, these differences were not significant in the logistic regression analysis. In the present study, we evaluated these markers and obtained similar results. Interestingly, placental bulging, which has never been evaluated in this context before, was found to be a significant predictor of emergent delivery before 36 weeks GA, with an OR of 4.351 (95% CI 1.329–14.242, $P$=0.015). Placental bulge is the outpouching of the uterus containing the placenta and represents the absence of myometrial tissue to support the placenta. This can be observed on US and is highly predictive of placental increta or percreta in the presence of other PAS markers [20]. Placental bulging may also be a sign of uterine scar dehiscence [20]. A possible hypothesis is that weakened myometrial tissue may be vulnerable and progress to uterine contraction, abdominal pain, and vaginal bleeding, similar to uterine dehiscence or rupture, thus necessitating emergent delivery in these cases.

Women who underwent emergent delivery before 36 weeks GA experienced significantly more blood loss than those who delivered after 36 weeks GA; however, other outcomes, such as organ injuries and ICU admission, were comparable. No maternal death occurred in either group, which likely resulted from the fact that our center has established a center of excellence over time for PAS, which comprises a skilled multidisciplinary team for PAS management. In contrast, multiple neonatal complications were significantly more common in patients who delivered emergently before 36 weeks. These outcomes were expected because the newborns were delivered at an earlier GA. Our findings underscore the importance of delaying delivery in selected cases to minimize neonatal morbidities due to premature birth.

The exclusion of women who underwent elective delivery between 34 and 35+6 weeks GA may introduce some biases as it was impossible to determine retrospectively whether, if not electively delivered at 34–36 weeks, these women would fall into the group that could continue beyond 36 weeks or would require emergent delivery before 36 weeks. PUC and APH were observed in this group of women more frequently than in those who delivered after 36 weeks, but less frequently than in those

who required emergent delivery. Severe forms of PAS (increta and percreta) were also more common in this group than in those who delivered after 36 weeks, almost similar to the group who delivered emergently before 36 weeks. These factors were likely the reasons for attending physicians to plan delivery early. Maternal outcomes, specifically EBL, were comparable between women who underwent elective delivery and those who delivered after 36 weeks. However, the composite neonatal outcomes in the elective group were similar to those in the group that underwent emergent delivery before 36 weeks and were much higher than those in the group that delivered after 36 weeks. This can also be explained by the earlier GA at delivery.

The strength of this study is the relatively large sample size of women prenatally diagnosed with PAS alone, not with placenta previa. We assessed the clinical factors at a specific GA from 14 to 34 weeks, the time at which the decision must be made, and whether to extend the pregnancy. This is also the first study to evaluate the risk factors for emergent delivery before 36 weeks GA, whereas previous studies only assessed the risk factors for emergent delivery at any gestational age. We believe that our study design and findings will be useful in clinical practice when dealing with women with PAS at approximately 34 weeks GA to decide on the timing of delivery.

This study has some other limitations that are worth mentioning. First, the retrospective nature of the study means that the quality of the data may be affected by the quality of the medical documentation. In addition, there were no rigid criteria before the decision was made for each patient to undergo emergent delivery; therefore, the threshold to deliver may vary among different physicians.

## Conclusions

In the current study, we identified the risk factors associated with emergent delivery at <36 weeks GA in women with PAS. Some of these have already been recognized, including a history of preterm birth in a prior pregnancy, PUC before 34 weeks GA, APH before 34 weeks GA, and a more severe form of PAS. A novel risk factor for placental bulging has also been reported. However, as each of these factors has unequal effects on patients, further prospective studies aiming to develop a scoring system with a cutoff value may be useful for decision-making in clinical practice.

## Supporting information

**S1 File. Supplementary data.**

(XLSX)

## Acknowledgments

We would like to thank Dr. Alan Geater, Epidemiology Unit, Faculty of Medicine, Prince of Songkla University, for his assistance with the statistical analysis and valuable comments.

## Author contributions

**Conceptualization:** Nawaporn Phetrat, Savitree Pranpanus, Thitima Suntharasaj, Chusana Petpichetchian.

**Data curation:** Nawaporn Phetrat, Savitree Pranpanus, Chusana Petpichetchian.

**Formal analysis:** Nawaporn Phetrat, Chusana Petpichetchian.

**Methodology:** Nawaporn Phetrat, Thitima Suntharasaj, Chusana Petpichetchian.

**Project administration:** Nawaporn Phetrat.

**Resources:** Savitree Pranpanus.

**Supervision:** Savitree Pranpanus, Thitima Suntharasaj, Chusana Petpichetchian.

**Writing – original draft:** Nawaporn Phetrat.

**Writing – review & editing:** Savitree Pranpanus, Thitima Suntharasaj, Chusana Petpichetchian.

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
