## [Decision Letter · Decision Letter 0]

20 Nov 2024

PONE-D-24-17196Risk factors for emergent delivery before 36 weeks among pregnant women with placenta accreta spectrum disorderPLOS ONE

Dear Dr. Petpichetchian,

Thank you for submitting your manuscript to PLOS ONE. After careful consideration, we feel that it has merit but does not fully meet PLOS ONE’s publication criteria as it currently stands. Therefore, we invite you to submit a revised version of the manuscript that addresses the points raised during the review process.

Reviewers appreciated the manuscript. However, some concerns are present. The main point regards adopted inclusion and exclusion criteria. The exclusion of cases who underwent elective CS before 36 weeks removed a population that may have worse characteristics from the study. A description of these cases and a comparison with the included population may be helpful. Second, a better description of methods, particularly US evaluation, is needed. 

We look forward to receiving your revised manuscript.

Kind regards,

Simone Garzon

Academic Editor

PLOS ONE

Journal Requirements:

Additional Editor Comments:

Reviewers appreciated the manuscript. However, some concerns are present. The main point regards adopted inclusion and exclusion criteria. The exclusion of cases who underwent elective CS before 36 weeks removed a population that may have worse characteristics from the study. A description of these cases and a comparison with the included population may be helpful. Second, a better description of methods, particularly US evaluation, is needed.

• I would suggest a language revision to improve some typos and grammatical errors.

• The introduction is well-written and clearly supports the study rationale.

• Methods. I would suggest providing some examples of conditions unrelated to PAS.

• Methods. Why did the authors exclude women who underwent planned delivery between 34-36 weeks? May this group represent patients without unplanned cesarean section?

• How was the list of patients retrieved, and eligible patients identified? Was a register of PAS cases present?

• Lines 135-140 highlight the cut-off of 36 weeks as the limit considered appropriate for all pregnancies with PAS. This questions why some patients underwent planned CS between 34-36 weeks. Because these patients were excluded, more severe cases would have been removed and representativeness reduced. I would suggest discussing this limitation of the study.

• Which is the source of data presented in lines 144-146?

• Lines 133-135. I would suggest clarifying the results of this evaluation. How were cases classified, and based on which criteria?

• Multivariable NOT multivariate

• Results. Lines 154-156. I would suggest providing some details regarding patients who underwent CS between 34-36 weeks and comparing them with the study population.

• Lines 166-167. Did the authors observe differences in the concordance rate based on the type of placental pathology?

• Table 1. What do the authors mean by the Number of PUCs? Number of episodes?

• Table 2. The Placenta previa is missing.

• I would suggest providing the definition for all US parameters in text or supplementary material to allow the study to be reproduced.

• Comparing outcomes between those who delivered before and after 36 weeks, considering that before 36 weeks, we only have emergent vs. both emergent and elective CS after, may introduce some biases.

• Providing the reasons for emergent CS of included cases would help understand whether these cases were mandatory. Moreover, comparing cases of those who had undergone elective CS before 36 weeks would allow for a more complete view.

• Discussion. Supporting that provided results would help identify patients who may benefit from an early elective CS around 34 weeks and require all emergent CS after 34 weeks. What is the mean gestational age at emergent CS?

Reviewers' comments:

Reviewer's Responses to Questions

**Comments to the Author**

1. Is the manuscript technically sound, and do the data support the conclusions?

Reviewer #1: Yes

Reviewer #2: Yes

2. Has the statistical analysis been performed appropriately and rigorously? 

Reviewer #1: Yes

Reviewer #2: Yes

3. Have the authors made all data underlying the findings in their manuscript fully available?

Reviewer #1: Yes

Reviewer #2: Yes

4. Is the manuscript presented in an intelligible fashion and written in standard English?

Reviewer #1: Yes

Reviewer #2: Yes

5. Review Comments to the Author

Reviewer #1: Thank you for a well written and researched article. However, I have a few minor corrections for better clarity

a. Page 10, lines 81-82: Cancel ‘s recent committee opinion’ as it is a repetition

b. Page 10, line 93: Cancel ‘and’

c. Page 11, lines 106-108, please clarify; Not clear

“, where delivery timing must be planned at approximately 34 weeks of DA to extend the pregnancy beyond 36 weeks

Reviewer #2: his is an extremely interesting retrospective study on the timing of delivery in pregnant women with placental abnormalities. The text is written in proper English, the statistical analysis appears methodologically sound, and the data presented in the results are consistent with the discussion. Overall, the study highlights possible risk factors that increase the likelihood of emergent preterm delivery in the patient category under study. It would be valuable to validate these findings by conducting a prospective study, which could help reduce the risk of bias.

6. PLOS authors have the option to publish the peer review history of their article (what does this mean? ). If published, this will include your full peer review and any attached files.

**Do you want your identity to be public for this peer review?** For information about this choice, including consent withdrawal, please see our Privacy Policy .

Reviewer #1: No

Reviewer #2: No

---

## [Author Response · Author response to Decision Letter 0]

17 Dec 2024

Additional Editor Comments:

1. I would suggest a language revision to improve some typos and grammatical errors.

A professional editing service has been used to ensure the grammatical accuracy and readability of the manuscript.

2. The introduction is well-written and clearly supports the study rationale.

Thank you for your kind comment.

3 Methods. I would suggest providing some examples of conditions unrelated to PAS.

Examples have been added to the Methods section on page 5, ‘such as maternal hypertensive disorders or fetal growth restriction’.

4. Methods. Why did the authors exclude women who underwent planned delivery between 34-36 weeks? May this group represent patients without unplanned cesarean section?

As our research question was to determine factors that would help select women who would benefit from pregnancy continuation beyond 36 weeks, we thought it would be best to exclude this group of women because it was impossible to retrospectively determine that if these women did not undergo elective delivery at 34-36 weeks, whether they would fall into the group that could continue beyond 36 weeks or would require emergent delivery before 36 weeks.

5. How was the list of patients retrieved, and eligible patients identified? Was a register of PAS cases present?

As our institution is the sole referral center for PAS in the region, we set up a registry for PAS cases in 2016. Cases of placenta previa and PAS have been prospectively collected since 2016. The list of women with diagnoses before that time was retrieved and also added to the registry from the hospital’s electronic database. The primary investigator for this study then manually selected cases from the registry.

6. Lines 135-140 highlight the cut-off of 36 weeks as the limit considered appropriate for all pregnancies with PAS. This questions why some patients underwent planned CS between 34-36 weeks. Because these patients were excluded, more severe cases would have been removed and representativeness reduced. I would suggest discussing this limitation of the study.

The data regarding this group of women has been added in the Result section, and additional discussion has been provided.

7. Which is the source of data presented in lines 144-146?

The data were from reference number 6 ‘Morlando M, Schwickert A, Stefanovic V, Gziri MM, Pateisky P, Chalubinski KM, et al. Maternal and neonatal outcomes in planned versus emergency cesarean delivery for placenta accreta spectrum: a multinational database study. Acta Obstet Gynecol Scand. 2021;100 Suppl 1: 41-49. doi: 10.1111/aogs.14120’

8. Lines 133-135. I would suggest clarifying the results of this evaluation. How were cases classified, and based on which criteria?

The diagnosis of PAS by USG was based on the descriptors proposed by the European Working Group on Abnormally Invasive Placenta. This clarification, along with the reference (no.19), has been added to the manuscript.

9. Multivariable NOT multivariate

Corrections have been made on pages 2 and 14

10. Results. Lines 154-156. I would suggest providing some details regarding patients who underwent CS between 34-36 weeks and comparing them with the study population.

Additional details about the group of patients are provided in the last paragraph of the Result section.

11. Lines 166-167. Did the authors observe differences in the concordance rate based on the type of placental pathology?

Yes, and the details were added into the text as follows.

A diagnosis of PAS was histologically confirmed in 150 patients (86.2%) and was more frequent in women with a prenatal suspicion of placenta percreta (26/26 women, 100%) than in those with placenta increta (77/84 women, 91.7%) or accreta (47/64 women, 73.4%).

12. Table 1. What do the authors mean by the Number of PUCs? Number of episodes?

Yes. For clarity, the term in the table has been changed to ‘episodes of PUCs.’

13. Table 2. The Placenta previa is missing.

Yes. That is because we included only cases with suspected PAS. Therefore, cases with a prenatal diagnosis of placenta previa alone were excluded.

14. I would suggest providing the definition for all US parameters in text or supplementary material to allow the study to be reproduced.

The ultrasound descriptors are in accordance with the reference that was mentioned earlier.

15. Comparing outcomes between those who delivered before and after 36 weeks, considering that before 36 weeks, we only have emergent vs. both emergent and elective CS after, may introduce some biases.

Additional details and discussion about the group that delivered electively at 34-36 weeks regarding the delivery outcomes have been provided.

16. Providing the reasons for emergent CS of included cases would help understand whether these cases were mandatory. Moreover, comparing cases of those who had undergone elective CS before 36 weeks would allow for a more complete view.

Reasons for emergent delivery have been added to the Results section (lines 175-178)

Additional details and discussion regarding the delivery outcomes have been provided about the group that delivered electively at 34-36 weeks.

17. Discussion. Supporting that provided results would help identify patients who may benefit from an early elective CS around 34 weeks and require all emergent CS after 34 weeks. What is the mean gestational age at emergent CS?

This information has been added to the Results as follows,

‘Emergent delivery before 36 weeks of GA was required in 45 (25.0%) women (mean GA at delivery 32+6 weeks), whereas pregnancy continued beyond 36 weeks of GA in 129 (75.0%) women (mean GA at delivery 37 weeks).’

Reviewer #1:

Thank you for a well written and researched article. However, I have a few minor corrections for better clarity

18. Page 10, lines 81-82: Cancel ‘s recent committee opinion’ as it is a repetition

Thank you. The correction has been made.

19. Page 10, line 93: Cancel ‘and’

Thank you. The correction has been made.

20. Page 11, lines 106-108, please clarify; Not clear

“, where delivery timing must be planned at approximately 34 weeks of DA to extend the pregnancy beyond 36 weeks

The text has been changed to ‘…where delivery timing must be planned at approximately 34 weeks GA, regarding whether to perform elective delivery before 36 weeks or extend the pregnancy beyond 36 weeks.’

Reviewer #2: his is an extremely interesting retrospective study on the timing of delivery in pregnant women with placental abnormalities. The text is written in proper English, the statistical analysis appears methodologically sound, and the data presented in the results are consistent with the discussion. Overall, the study highlights possible risk factors that increase the likelihood of emergent preterm delivery in the patient category under study. It would be valuable to validate these findings by conducting a prospective study, which could help reduce the risk of bias.

Thank you for your positive comments.

---

## [Decision Letter · Decision Letter 1]

9 Mar 2025

Risk factors for emergent delivery before 36 weeks among pregnant women with placenta accreta spectrum disorder

PONE-D-24-17196R1

Dear Dr. Petpichetchian,

We’re pleased to inform you that your manuscript has been judged scientifically suitable for publication and will be formally accepted for publication once it meets all outstanding technical requirements.

Kind regards,

Simone Garzon

Academic Editor

PLOS ONE

Additional Editor Comments (optional):

The manuscript and the reviewers’ comments were carefully evaluated. The Reviewers appreciated the manuscript and the improvements. Now it can be accepted. Compliments!

Reviewers' comments:

Reviewer's Responses to Questions

**Comments to the Author**

1. If the authors have adequately addressed your comments raised in a previous round of review and you feel that this manuscript is now acceptable for publication, you may indicate that here to bypass the “Comments to the Author” section, enter your conflict of interest statement in the “Confidential to Editor” section, and submit your "Accept" recommendation.

Reviewer #2: All comments have been addressed

2. Is the manuscript technically sound, and do the data support the conclusions?

Reviewer #2: Yes

3. Has the statistical analysis been performed appropriately and rigorously? 

Reviewer #2: Yes

4. Have the authors made all data underlying the findings in their manuscript fully available?

Reviewer #2: Yes

5. Is the manuscript presented in an intelligible fashion and written in standard English?

Reviewer #2: Yes

6. Review Comments to the Author

Reviewer #2: I believe that the author has made the necessary revisions to the manuscript in accordance with the feedback and recommendations provided by both the editor and the reviewers. I highly appreciate the effort put into addressing these points, and I find the revised version to be significantly improved and more aligned with the required standards.

7. PLOS authors have the option to publish the peer review history of their article (what does this mean? ). If published, this will include your full peer review and any attached files.

**Do you want your identity to be public for this peer review?** For information about this choice, including consent withdrawal, please see our Privacy Policy .

Reviewer #2: No

---

## [Editor Report · Acceptance letter]

PONE-D-24-17196R1

PLOS ONE

Dear Dr. Petpichetchian,

I'm pleased to inform you that your manuscript has been deemed suitable for publication in PLOS ONE. Congratulations! Your manuscript is now being handed over to our production team.

Kind regards,

on behalf of

Dr. Simone Garzon

Academic Editor

PLOS ONE